# Immunophenotypic Analysis of Hairy Cell Leukemia (HCL) and Hairy Cell Leukemia-like (HCL-like) Disorders

**DOI:** 10.3390/cancers14041050

**Published:** 2022-02-18

**Authors:** Elsa Maitre, Edouard Cornet, Véronique Salaün, Pauline Kerneves, Stéphane Chèze, Yohan Repesse, Gandhi Damaj, Xavier Troussard

**Affiliations:** 1INSERM1245, MICAH, Normandie University of Caen and Rouen, UNIROUEN, UNICAEN, Avenue de la Côte de Nacre, 14033 Caen, France; maitre-e@chu-caen.fr (E.M.); damaj-gl@chu-caen.fr (G.D.); 2Laboratory of Hematology, University Hospital Caen, Avenue de la Côte de Nacre, 14033 Caen, France; cornet-e@chu-caen.fr (E.C.); salaun-v@chu-caen.fr (V.S.); kerneves-p@chu-caen.fr (P.K.); repesse-y@chu-caen.fr (Y.R.); 3Hematology Institute, University Hospital Caen, Avenue de la Côte de Nacre, 14033 Caen, France; cheze-s@chu-caen.fr

**Keywords:** hairy cell leukemia (HCL), hairy cell leukemia-like disorders, hairy cell leukemia variant (vHCL), splenic diffuse red pulp lymphoma, flow cytometry

## Abstract

**Simple Summary:**

Hairy cell leukemia (HCL) is a rare B cell neoplasm that accounts for 2% of B-cell lymphomas. The diagnosis was based on the presence of abnormal lymphoid cells that expressed CD103, CD123, CD25 and CD11c. The aim of this retrospective study was to describe the immunophenotypic profile of HCL and HCL-like disorders using 13 markers and to assess the added value of immunophenotypic row data and unsupervised analysis. We confirmed that the immunological profile alone is not sufficient and that morphologic, phenotypic and molecular data need to be integrated.

**Abstract:**

Hairy cell leukemia (HCL) is characterized by abnormal villous lymphoid cells that express CD103, CD123, CD25 and CD11c. HCL-like disorders, including hairy cell leukemia variant (vHCL) and splenic diffuse red pulp lymphoma (SDRPL), have similar morphologic criteria and a distinct phenotypic and genetic profile. We investigated the immunophenotypic features of a large cohort of 82 patients: 68 classical HCL, 5 vHCL/SDRPL and 9 HCL-like NOS. The HCL immunophenotype was heterogeneous: positive CD5 expression in 7/68 (10%), CD10 in 12/68 (18%), CD38 in 24/67 (36%), CD23 in 22/68 (32%) and CD43 in 19/65 (31%) patients. CD26 was expressed in 35/36 (97%) of HCL patients, none of vHCL/SDRPL and one of seven HCL-like NOS (14%). When adding CD26 to the immunologic HCL scoring system (one point for CD103, CD123, CD25, CD11c and CD26), the specificity was improved, increasing from 78.6% to 100%. We used unsupervised analysis of flow cytometry raw data (median fluorescence, percentage of expression) and the mutational profile of *BRAF, MAP2K1* and *KLF2*. The analysis showed good separation between HCL and vHCL/SDRPL. The HCL score is not sufficient, and the use of unsupervised analysis could be promising to achieve a distinction between HCL and HCL-like disorders. However, these preliminary results have to be confirmed in a further study with a higher number of patients.

## 1. Introduction

Hairy cell leukemia (HCL) and HCL-like disorders, including hairy cell leukemia variant (vHCL), splenic diffuse red pulp lymphoma (SDRPL) and splenic marginal zone lymphoma (SMZL) with villous lymphoid cells, are characterized by a similar morphologic examination, with the identification of hairy cells in the blood and bone marrow and a distinct phenotypic and genetic profile, a different clinical course and the need for appropriate treatment.

Initially described in 1958 [1], HCL is a well-defined and distinct entity in the 4th revised 2017 classification of the World Health Organization (WHO) of hematopoietic and lymphoid tumors [2]. With approximately 1500 new yearly cases in Europe and the United States, HCL is four to five times more common in men than women [3,4]. The median age of patients at diagnosis is 63 years in men and 59 years in women [4]. Patients present at diagnosis with splenomegaly, pancytopenia or infections. The diagnosis is based on the identification of characteristic hairy cells involving peripheral blood and diffusely infiltrating the bone marrow and the splenic red pulp. The *BRAF-V600E* mutation in the B-raf proto-oncogene (*BRAF* gene) (7q34), which is detected in 80–90% of HCL cases, is also identified in various solid tumors such as melanoma [5]. The mutation mimics phosphorylation independently from RAS, resulting in constitutive kinase activity. Downstream kinases activation by mitogen-activated protein kinase kinases (MEKs) MEK1 and MEK2 leads to phosphorylate extracellular signal-regulated kinases (ERKs) ERK1 and ERK2. Ex vivo and human studies have shown that HCL cells present high levels of MEK and ERK phosphorylation with a reduction in levels induced by inhibitors of BRAF (BRAFi), such as vemurafenib or dabrafenib, suggesting that aberrant signaling through the BRAF–MEK–ERK pathway could be an ideal therapeutic target in HCL [6,7]. Weston-Bell et al. [8] also reported that B-cell receptors (BCRs) of HCL cells respond to antibody-mediated cross-linking with an increase in cellular calcium levels, ERK phosphorylation, and apoptosis. In contrast, the ability of BCR cross-linking to protect primary HCL cells from undergoing spontaneous apoptosis was reported in vitro; pretreatment with the BTK inhibitor (BTKi) ibrutinib completely abrogated these effects, suggesting the therapeutic relevance of BTKi, particularly in HCL patients [8,9,10].

HCL could be confused with other HCL-like disorders, including vHCL and SDRPL. To distinguish the different entities, flow cytometry was used: an immunological scoring system was developed with one point given to each of the four (CD11c, CD25, CD103, CD123) markers when it was expressed and no point when it was not expressed. A score of 3 or 4 is observed in 98% of cases of HCL, whereas the score is low and <3 in other HCL-like disorders [11]. CD200, which is brightly expressed, could be useful for HCL diagnosis [12,13].

vHCL is a provisional entity, representing 800 new incident cases in the United States [3]. Circulating abnormal lymphoid cells present hybrid criteria between prolymphocytes and hairy cells. The immunological score is low: there is usually no expression of CD25 and CD200. CD123 expression is inconstant and weak [14]. A high prevalence of activating mutations in the mitogen-activated protein kinase 1 (MAP2K1) gene (15q22.1–q22.3) encoding MEK1 was identified with an overall frequency of 48%. *MAP2K1* mutations are predominantly detected in HCL patients expressing *IGHV4-34* and in vHCL regardless of the IGHV rearrangements [15]. *TP53* aberrations, accounting for 30–40% of patients, are associated with a significant risk for chemoresistance [16].

SDRPL is also a provisional entity very close if not identical to vHCL [17]. A high proportion with a median of 60% of small- to medium-sized villous lymphoid cells is present in the peripheral blood; the cells have a polar distribution of the villi, and the nucleolus is small or not visible. Monoclonal B cells express CD11c (97%), inconsistently CD103 (38%) and rarely CD123 (16%) or CD25 (3%) [18]. The CD200/CD180 median fluorescence (MFI) ratio may be helpful to distinguish HCL from SDRPL, with a ratio of 0.5 or less in favor of SDRPL [12]. As in vHCL, a *BRAF-V600E* mutation was never detected. *CCND3* mutations involving the regulatory PEST domain and leading to cyclin D3 overexpression were observed in less than 25% of patients [19,20].

SMZL with villous lymphocytes is characterized by the presence of abnormal lymphoid cells with round nuclei, condensed chromatin, and basophilic cytoplasm with polar short villi in the peripheral blood. Heterogeneity in blood morphology is common, ranging from small lymphoid cells without specific features to various degrees of monocytoid and plasmacytoid differentiation. A scoring system based on CD11c, CD22, CD76, CD38 and CD27 expression was designed to differentiate SDRPL from SMZL [18]. Unlike HCL and vHCL, where the red pulp of the spleen is infiltrated, SMZL develops in the white pulp with a biphasic picture; lymphoma cells may involve the red pulp in a patchy or diffuse fashion with subsequent spread to the sinuses.

Patients with HCL and vHCL should be treated only if they are symptomatic [21]. Chemotherapy with purine analogs (PNAs) is indicated in the first line. The use of chemoimmunotherapy combining PNAs with rituximab (R) represents an increasingly used front-line therapeutic approach. Long remissions are typically achieved, but most cases relapse and require further therapy. In relapsed/refractory HCL patients, novel therapeutic approaches are needed to utilize less immunosuppressive drugs than PNAs. Because of a different clinical course and the need for appropriate treatment, it is necessary to distinguish the different entities and better classify patients. In this article, we investigated a large series of 82 patients with HCL or HCL-like disorders in a single center by flow cytometric immunophenotyping (FCI).

## 2. Materials and Methods

Eighty-two patients were studied between July 2003 and June 2021. Of the 102 samples analyzed, 55 were at diagnosis and 47 at relapse. The samples were in most cases peripheral blood (*n =* 65), medullary (*n* = 36) or splenic (*n* = 1). The diagnosis was based on the WHO 2017 classification [2]. The peripheral blood and/or medullary morphologic smears of all 82 cases were reviewed. Three groups of patients were identified: group 1 (cHCL) included patients with a typical HCL morphology and an immunologic score ≥ 3, and group 2 (vHCL/SDRPL) included patients with either vHCL or SDRPL diagnosis and an immunologic score ≤ 3. Because of an overlap between SMZL, SDRPL and vHCL, some patients with abnormal and villous lymphoid cells in the peripheral blood were classified into HCL-like NOS group 3, requiring additional analysis (IGHV profile, mutational landscape, etc.).

Flow cytometry immunophenotyping (FCI). Multiparameter flow cytometric immunophenotyping performed on a FACSCANTO II or a FACSCalibur (Becton Dickinson, San Diego, CA, USA) was used to characterize hairy cells (HCs) and quantify tumor infiltration. PBMCs (5 × 10^5^ cells) were incubated for 30 min at 4 °C with the following antibodies: anti-CD45-V450, -CD5-PerCpCy5,5, -CD19-PECy7, -CD23-PE, -CD43-FITC, -CD10-APC, -CD38-V450, -CD103-FITC, -CD123-PE, -CD11c-V421, -CD25-APC, CD26-PE, CD27-PercpCy5,5, -CD4-PerCpCy5,5, and -CD8-APC-H7 purchased from Becton Dickinson or anti-κ-FITC and -λ-PE from Dako (Dako Corporation, Carpintero, CA, USA). Antibody excess was washed with the lyse/lavage BD FACS^®^ device (Becton Dickinson, CA, USA). Devices were standardized with FranceFlow recommendations. Data were analyzed using FACSDiva software version 8 (Becton Dickinson, San Diego, CA, USA). Positivity is defined by ≥20% of abnormal cells expressing the marker. Diminished (dim) expression was defined by MFI < 500, and bright expression was defined by MFI > 1000. A ratio of CD4/TCD8 T-cells with a cut off ratio >2 was used in the survival analysis. Percentage of tumor infiltration was defined by the percentage of abnormal cells among the total lymphocytes.

Genetic profile. DNA was extracted with the automated MagnaPur^®^ device (Roche Life Science, Rotkreuz, Swizerland) according to the manufacturer’s recommendations. Sequencing of *BRAF* (exons 11, 15), *MAP2K1* (exons 2–3) and Krüppel-like 2 (*KLF2)* (exons 1–3) was performed on Ion Torrent S5/PGM^TM^ devices as previously described [22].

Data representation and statistical analysis. Data representations were made using R (version 4.4.1: https://www.R-project.org/, accessed on 25 October 2021). Principal component analysis was performed with FactomineR packages [23]. Statistical representations of the Kaplan–Meier test on time to next treatment (TTNT), progression-free survival (PFS) and overall survival (OS) were performed using the survminer package [24], and *p* values were calculated with the log-rank test. TTNT was calculated from the ending date of the first treatment to the date of second treatment or last patient follow-up. OS was calculated from the date of diagnosis to the date of death or last patient follow-up. PFS was calculated from the date of diagnosis until disease progression, relapse, death or last patient follow-up. *p* values < 0.05 were considered statistically significant.

## 3. Results

Among the 82 patients, 63 were men and 19 were women, with ages ranging from 30 to 95 years at diagnosis and a mean age of 59.6 years. Sixty-eight patients (83%) were classified into group 1 (cHCL), 9 (11%) into group 3 (HCL-like NOS) and 5 (6%) into group 2 (vHCL/SDRPL). The clinical and biological characteristics of the whole cohort and the different groups are listed in Appendix A. Among the 82 patients, 78 showed surface immunoglobulin light chain restriction, either kappa (*n =* 47) or lambda (*n* = 31), and were positive for the pan-B-cell markers CD19 and CD20. The other four remaining patients had undetectable surface immunoglobulin light chains.

In group 1, the immunologic score was four in 65/68 (96%) cHCL and three in three cases (4%). CD11c was bright in all cases, and the MFI was higher in HCL than in HCL-like NOS or vHCL/SDRPL (Appendix A, Figure 1). CD123 was expressed in 60/62 (97%) patients, with only two patients who did not express CD123. CD103 was homogenously expressed in 66/67 (99%) patients, and CD25 was positive in all patients. BRAF-V600E was identified in 46/52 HCL cases (88%). BRAF-V600E or other alternative BRAF mutations were not detected in six patients.

In HCL-like disorders (including groups 2 and 3 as defined in the Materials and Methods), the score was <3 in 11/15 (73%) patients, 4/5 vHCL/SDRPL (80%) and 7/9 HCL-like NOS (78%) (Appendix A). CD25 was expressed in 1/5 of patients with vHCL/SDRPL (20%) and 2/9 of patients (22%) with HCL-like NOS. CD123 was expressed in 1/5 (20%) vHCL/SDRPL and 2/9 (22%) HCL-like NOS patients. CD103 was frequently expressed in 4/5 (80%) and 3/9 (33%) HCL-like NOS patients. *BRAF* was wild type in all HCL-like NOS and vHCL/SDRPL.

In this large cohort, an unusual phenotype was detected in 55 patients. Among HCL patients, some markers were heterogeneously expressed (Appendix A, Figure 1). CD5 and CD10 expression was identified in 7/68 (10%) and 12/68 (18%) cases, respectively. CD38, CD23 or CD43 expression varied between bright and dim in 24 (36%), 22 (32%) and 19 (31%) patients tested, respectively. The analysis of cHCL depending on the expression of CD43, CD5, CD10, CD38 and CD4/CD8 T cells ratio did not show a significant difference in OS or PFS but a longer TTNT in the case of CD23-positive expression (144.4 vs. 63.1 months) (Figure 2a and Appendix A), a difference that was not found to be significant (*p* = 0.07). The ratio of CD4/CD8 T cells has a significant clinical impact, with a better prognosis in patients with a CD4/CD8 ratio > 2 and a longer TTNT compared with that observed in patients with a ratio < 2 (174.7 versus 67.8 months) (Figure 2b).

CD26 was positive in 35/36 cHCL cases (97%), including two cHCL *BRAF^WT^* (UPN-75, UPN-91) and CD27, a marker of memory B cells, was expressed in 11/46 cHCL samples that were tested (24%). CD27 was positive in 2/5 (40%) and 2/7 (29%) vHC/SDRPL and HCL-like NOS, respectively, and CD26 was negative in 2/2 vHCL/SDRPL and 6/7 HCL-like-NOS samples. When adding CD26 to the immunologic HCL scoring system, the specificity was improved, increasing from 78.6% to 100%, and the sensitivity remained stable at 100% (Appendix A).

### Unsupervised Analysis

Because overlap does exist in the score between cHCL and HCL-like disorders, we used unsupervised analysis on raw data (% of expression and median of fluorescence). Unsupervised analysis by principal component analysis (PCA) allowed the use of multiple continuous variables and reduced it to two dimensions to visualize samples associated with clusters. For this analysis, we used 86 samples analyzed on the same FACS CANTO-II device (Becton Dickinson), including cHCL (*n* = 72), vHCL/SDRPL (*n* = 5) and HCL-like NOS (*n* = 9). Using all 13 immunophenotypic markers (MFI, %), in addition to the percentage of tumor infiltration, PCA allowed good discrimination between cHCL and vHCL/SDRPL (Figure 3a). The immunophenotype of HCL-like NOS was closer to vHCL/SDRPL than that identified in cHCL. However, there was a gray zone in eight cHCL samples and HCL-like NOS samples, which could be explained by the lack of CD123 expression in UPN-51 and UPN-46 and a higher tumor infiltration (54.6% vs. 17.1% *p* = 6.047 × 10^−5^) (Appendix A). In this PCA, patients with an HCL score of 3 presented an immunophenotypic profile close to either a score of 4 or 1 (Figure 3b). The FCI was not correlated with the *BRAF* mutational status (Figure 3c). To improve PCA clustering, immunophenotypic markers were combined with genetic features (e.g., mutational status of *BRAF*, *MAP2K1* and *KLF2*), allowing higher variance (Figure 4). When including all these parameters, the distinction between cHCL and vHCL/SDRPL was more distinct. Patients classified as cHCL with *BRAF^WT^* were split into two subgroups. The first was close to HCL (UPN-10, UPN-91, UPN-75, UPN-55), and the second was closer to vHCL/SDRPL (UPN-40, UPN-46). Two other patients presented *MAP2K1* mutations and an unmutated *IGHV VH4-34* profile. UPN-v18, initially classified in HCL-like NOS because of the absence of CD25 expression, was reclassified by unsupervised analysis in the cHCL group (Figure 4a). Among the eight cHCL patients in the gray area between cHCL and HCL-like NOS by using FCI, three patients were reclassified with the addition of the genetic profile, the other remaining patients being closer to cHCL, allowing them to be removed from the gray zone. Finally, all the patients were classified into two groups according to PCA clustering: HCL and HCL-like (Figure 4b).

Analysis of blood cell count is useful for HCL classification. When adding the blood cell count (absolute lymphocyte count (ALC), absolute monocyte count (AMC), absolute neutrophil count (ANC), platelet count (PC), hemoglobin (Hb), CD4/CD8 T cell ratio and CD26 expression data, PCA achieved good discrimination between classical HCL and HCL-like disorders without overlap (Appendix A).

## 4. Discussion

Because of the different clinical courses between patients and relevant therapeutic implications, FCI is useful for distinguishing HCL and HCL-like disorders. The utility of the immunologic HCL score is still relevant: the expression of CD103, CD123, CD25 and CD11c was described in more than 95% of cases [11,14].

Overexpression of integrin proteins (CD11c and CD103), interleukin receptors to IL-2 (CD25) and IL-3 (CD123) was correlated with high mRNA levels [25]. Transcriptomic studies also showed that hairy cells were derived from memory B cells, despite the absence of CD27 expression [25]. Unlike a few studies where CD27 was never expressed in HCL [26], 24% of cHCL samples (11/46) and 40% of 2/5 vHCL/SDRPL patients were CD27 positive in our study. Unusual phenotypes were previously published, and CD5 and CD10 expression was rare and positive in less than 10% and 10–20% of cases, respectively [27,28,29,30]. In our cohort, CD5 expression was higher (7/68, 10%) than usually described, which could be due to differences in experimental conditions. CD5 expression was performed in the same tube as the kappa and lambda light chain, allowing a good sensitivity. Because of the small number of vHCL, we were not able to confirm CD79b and CD43 overexpression in HCL compared with vHCL/SDRPL [31], but in contrast to HCL, 2/5 vHCL/SDRPL cases did not express CD79b. CD23 expression was initially reported as negative in HCL and subsequently positive in 17% of cases [27]. In our study, the percentage of CD23 expression was slightly higher in 33% of cHCL cases and correlated with a good prognosis with a longer TTNT. In our study, CD38 expression was not associated with a shorter TTNT and could be explained by the small size of the cohort and the type of treatment used [32].

The tumor microenvironment is crucial in HCL pathogenesis because of the expression of multiple adhesion proteins (CD11c, CD103, VLA-4) and the induction of bone marrow fibrosis (bFGF) [33]. Programmed death ligand 1 (PDL-1) is highly expressed in HCL and could be modulated by tumor-infiltrating T cells [34], and the increase in CD8 infiltration was correlated with a durable 2-CDA response [35]. In our study, a normal CD4/CD8 T cell ratio (>2) was associated with a longer TTNT.

An immunologic HCL score >3 is highly correlated with HCL, and in the initial publication, no vHCL had a score of 3 because of the absence of CD25 and CD123 expression [11]. In a more recent series, a score of 3 was described in vHCL cases with dim CD25 and/or CD123 expression [36,37]. In our cohort, three cHCL and two HCL-like NOS patients had an HCL score of 3. The cause of this score varied (absence of CD123 expression in three cases, CD103 in two cases and CD25 in one case). CD26/DPPIV (dipeptidyl peptidase IV) has an essential role in immune regulation by activating T cells. CD26 is physically and functionally associated with CXCR4 (CD184), the receptor of stromal cell-derived factor 1 (CXCL12), which is involved in migration and adhesion. We showed in our study that CD26 was widely expressed in HCL [38] but absent from HCL-like disorders. When adding CD26 and modifying the HCL immunologic score system based here on five points, the classification was improved, reaching a sensitivity and specificity of 100%.

Because the HCL score does not reflect heterogeneity in expression (bright or dim and percentage of abnormal cells that expressed the marker), the use of raw flow cytometry data independent of the cut off could help to classify HCL diseases. Independent classification using immunophenotypic variables has been successfully used in mature B cell neoplasms [39], but our knowledge has not been tested to discriminate villous lymphoproliferative neoplasms. Unsupervised analysis based on all immunophenotypic markers allowed good discrimination between cHCL and vHCL/SDRPL independent of the immunologic HCL score and *BRAF* status. The eight cHCL patients who overlapped with HCL-like NOS had larger tumor infiltration and atypical immunophenotypic expression with the absence of CD123 in two cases.

To improve discrimination, the combination of a panel of 13 immunological markers and the mutated or unmutated status of three genes (*BRAF*, *KLF2*, *MAP2K1*) was more relevant and closer to clinical practice. The recurrent *BRAF^V600E^* mutation was described in 80–100% of HCL cases [5,40], and *MAP2K1* mutations were described in 50% of vHCL cases [15].

This PCA performed on limited but more relevant variables highlighted a potential misclassification of HCL subgroups, especially on the cHCL *BRAF^WT^* group. Indeed, patients classified in cHCL with *BRAF^WT^* split into two groups: one closer to cHCL (UPN-10, UPN-91, UPN-75, UPN-55) and the second closer to vHCL/SDRPL (UPN-40, UPN-46). When detailed, the two remaining patients presented *MAP2K1* mutations and an IGHV VH4-34 unmutated profile that is described in more aggressive disease and particularly in vHCL disease [15,37,41]. In contrast, one HCL-like NOS patient (UPN-v18) had an immunophenotypic and genotypic profile closer to cHCL than vHCL/SDRPL. Thus, unsupervised analysis allowed us to reclassify patients into two groups according to PCA clustering: HCL and HCL-like. Due to a low number of patients, it was not possible to split the cohort into a training and a validation set. The classification must be validated in a prospective cohort.

To further improve PCA, the addition of CD26 expression and blood count parameters was useful in discriminating between HCL and HCL-like disorders. The absolute lymphocyte count was higher in HCL-like disorders, and monocytopenia was more frequent in HCL [42]. Because the data were available in only 59 patients, a larger cohort will be needed to confirm the interest of these additional parameters.

## 5. Conclusions

The immunophenotypic expression of hairy cells is heterogeneous, and the HCL score is not sufficient to allow good classification. The addition of CD26 positivity in the scoring could be an interesting approach. Additionally, the use of raw data (% of expression and MFI) and unsupervised analysis have the advantage of avoiding interpretation and allowing more reproducible analysis. The maximum of variance was obtained with the use of limited immunophenotypic and genotypic combined markers (percentage of expression + median of fluorescence of CD11c, CD103, CD123, and CD25; percentage of abnormal cells on sample, *BRAF*, *MAP2K1*, and *KLF2* mutation status). CD26 usefulness and PCA classification must be validated in a prospective cohort.

## Figures and Tables

**Figure 1 cancers-14-01050-f001:**
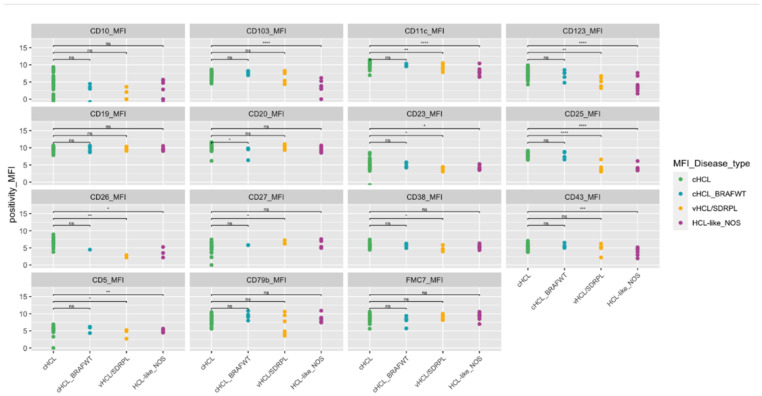
Phenotypic characteristics of the patients. Median fluorescence for each marker tested (cHCL *n* = 62, cHCL BRAFWT *n* = 5, vHCL/SDRPL *n* = 5, HCL-like NOS *n* = 9). Brown line underline 20%.Abbreviations: cHCL: classical form of Hairy cell leukemia, vHCL: variant form of hairy cell leukemia, SDRPL: splenic diffuse red pulp lymphoma, HCL-like NOS: hairy cell leukemia like disease not otherwise specified, ns: not significant, *: *p*-value ≤ 0.05, **: *p*-value ≤ 0.01, ***: *p*-value ≤ 0.001, ****: *p*-value ≤ 0.0001.

**Figure 2 cancers-14-01050-f002:**
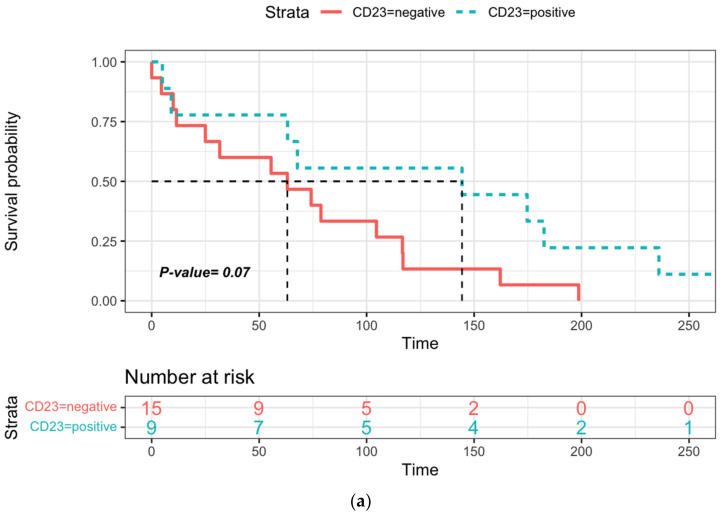
Time to next treatment (TTNT) in cHCL: (**a**) according to CD23 expression, (**b**) according to the CD4/CD8 T cell ratio (cut off CD4/TCD8 T cell ratio > 2). Dotted lines represent median of TTNT and PFS and red and blue areas the confidence intervals of survival curves.

**Figure 3 cancers-14-01050-f003:**
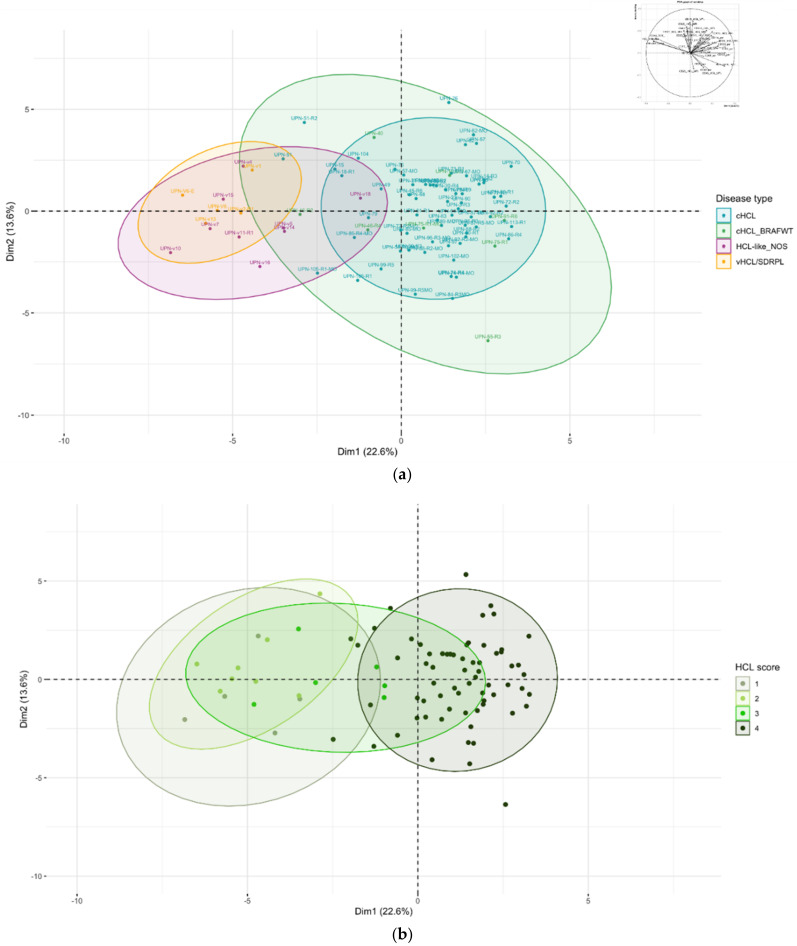
(**a**) Samples colored according to the disease; (**b**) samples colored according to the HCL score; (**c**) samples colored according to the BRAF mutational status. Component principal analysis (CPA) projection (first versus second principal component). Each dot corresponds to a sample (cHCL *n* = 64, cHCL BRAFWT *n* = 8, vHCL/SDRPL *n* = 5, HCL-like NOS *n* = 9). Variables used: (percentage of expression + median of fluorescence of CD19, CD20, CD79b, FMC7, CD5, CD10, CD23, CD43, CD38, CD11c, CD103, CD123, CD25; percentage of abnormal cells on sample, percentage of abnormal cells compared to total lymphocytes). Variables factor map in the right corner. Ellipses represent confidence level for a concentration of 90%.

**Figure 4 cancers-14-01050-f004:**
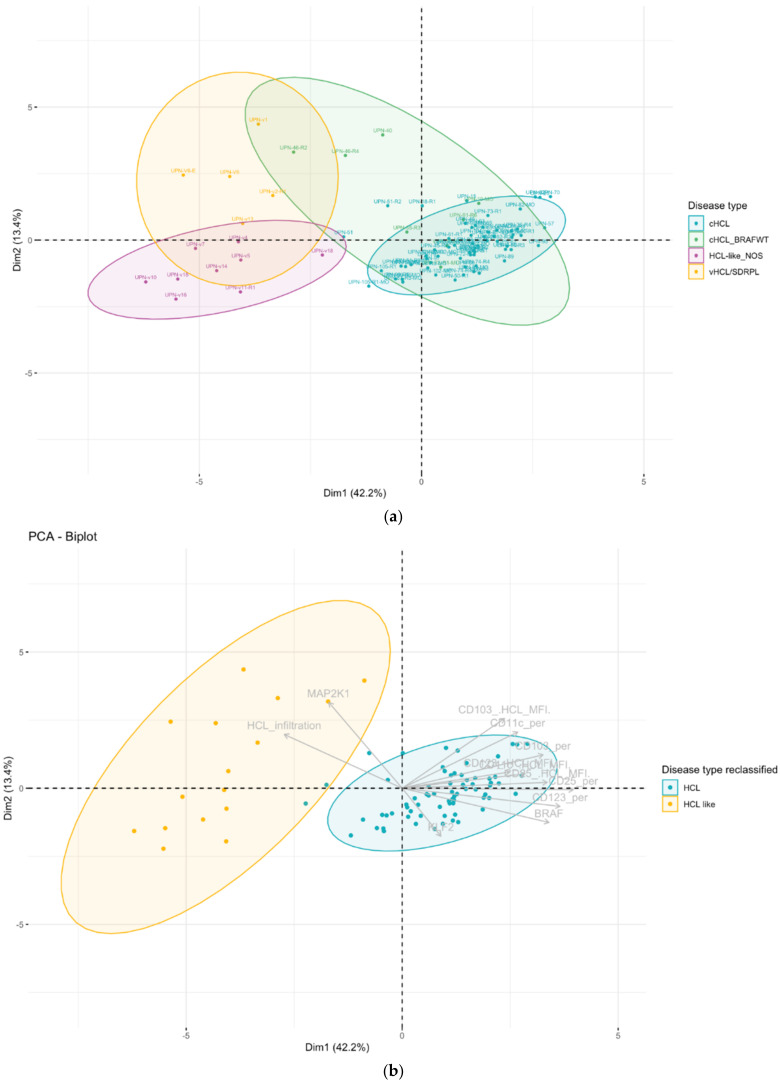
(**a**) Samples colored according to the disease; (**b**) samples colored according to reclassification disease type. In gray are the variables. Ellipses represent confidence level for a concentration of 90%. Component principal analysis (CPA) projection (first versus second principal component). Each dot corresponds to a sample (cHCL *n* = 64, cHCL BRAFWT *n* = 8, vHCL/SDRPL *n* = 5, HCL-like NOS *n* = 9). Variables used: (percentage of expression + median of fluorescence of CD11c, CD103, CD123, CD25; percentage of abnormal cells on sample, *BRAF*, *MAP2K1*, *KLF2* mutation status). Variable factors are shown in grey in the center. Ellipses represent confidence for a level concentration of 90%.

## Data Availability

Data available on request due to restrictions eg privacy or ethical The data presented in this study are available on request from the corresponding author.

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
