# Peer review of "Immunophenotypic Analysis of Hairy Cell Leukemia (HCL) and Hairy Cell Leukemia-like (HCL-like) Disorders"

_cancers, 2022, doi:10.3390/cancers14041050_

Round 1

Reviewer 1 Report

The authors attempt to separate hairy cell leukemia and hairy cell like disorders by immunophenotypic flow cytometric antigen expression with unsupervised analysis of findings.

Abstract:

Page 1, line 24. CD5 expression in your cases I higher than usually reported. Can you speculate for the reason?

Page 1, line 29. Should “row data” be “raw data”?

Introduction:

Page 1, Line 44. …revised 2016 classification of the World Health Organization. This should be 2017, the reference is correct.  

Results: 

Page 4, line 167. Is this group 2 or 3, please be specific.

Page 4, line 178 and Figure 2B. You introduce new immunophenotypic markers (CD4 and CD8) to determine “time to next treatment”. Please add this to the materials and methods.

3.1 Unsupervised Analysis.

Page 6. Line 205.

The authors introduce “percentage of tumor infiltration” as one of the variables used. Please add this to the materials and methods and what criteria were used.

Discussion:

Pate 11, line 291. …the use of row flow … should be “raw”.

Author Response

Reviewer 1

The authors attempt to separate hairy cell leukemia and hairy cell like disorders by immunophenotypic flow cytometric antigen expression with unsupervised analysis of findings.

Abstract:

Page 1, line 24. CD5 expression in your cases I higher than usually reported. Can you speculate for the reason?

Thank you for the remark. CD5 expression is variable in cHCL; it was described in 1/19 cases (5%) in Jain et al. 2016, 1/11 cases (9%) in Del Guidice et al. 2004, in 6/63 patients (10%) in Juliusson et al, 1994, and 4/169 patients (2%) in Shao et al 2013.

The differences could be explained by the analytical conditions and the way the panel was constructed. We used preferentially an 8-color flow cytometer, 500,000 PBMC were labelled and CD5 antibody was in the same tube as the kappa-lambda light chain, which allowed a very good sensitivity. We added and included the information in the manuscript (discussion: second paragraph).

Page 1, line 29. Should “row data” be “raw data”?

This a mistake: we modified as follows:

Line 29: flow cytometry raw data.

Introduction:

Page 1, Line 44. …revised 2016 classification of the World Health Organization. This should be 2017, the reference is correct.  

2016 was not correct. We modified. Line 44: in the 4th revised 2017 classification of the World Health Organization (WHO).

Results: 

Page 4, line 167. Is this group 2 or 3, please be specific.

Thank you for the comment. It is a good suggestion.

It was clarified as follows:

Line 167: In HCL-like disorders (including the groups 2 and 3 as defined in the materials and methods), the score was <3 in 11/15 (73.5%) patients: 4/5 vHCL/SDRPL (80%) and 7/9 HCL-like NOS (77.8%) (Table S1).

Page 4, line 178 and Figure 2B. You introduce new immunophenotypic markers (CD4 and CD8) to determine “time to next treatment”. Please add this to the materials and methods.

It was clarified and we changed (materials and methods: second paragraph)

Line 128: –CD4-PerCpCy5,5, -CD8-APC-H7 purchased from Becton Dickinson.

Line 134: a ratio of CD4/TCD8 T-cells with a cut off ratio >2 were used in the survival analysis.

3.1 Unsupervised Analysis.

Page 6. Line 205.

The authors introduce “percentage of tumor infiltration” as one of the variables used. Please add this to the materials and methods and what criteria were used.

Line 135: The percentage of tumor infiltration was defined by the percentage of abnormal cells among the total lymphocytes. We included the information.

Discussion:

Pate 11, line 291. …the use of row flow … should be “raw”.

It was corrected, as follows: Line 291: the use of raw flow cytometry data.

Reviewer 2 Report

According to this analysis the best approach to the diagnosis of cHCL is the use of limited immunophenotype, genotype and clinical combined markers. The limited number of patients in different subgroups probably prevent the definition of a diagnostic model with a score assigned to each marker. However, an effort in this direction could help clinicians to make a correct diagnosis of cHCL. 

Author Response

Reviewer 2

According to this analysis the best approach to the diagnosis of cHCL is the use of limited immunophenotype, genotype and clinical combined markers. The limited number of patients in different subgroups probably prevent the definition of a diagnostic model with a score assigned to each marker. However, an effort in this direction could help clinicians to make a correct diagnosis of cHCL. 

Many thanks for the comment.

A higher number of patients could be an advantage to improve the classification. Due to the rarity of the disease, the number of the patients is limited but reflects the real life.

Reviewer 3 Report

This an interesting paper worth publication.  The english can be improved.

For example this sentence (ase targets is activated: mitogen-activated protein ki- 54
nase kinases (MEKs) MEK1 and MEK2, which phosphorylate extracellular signal-regu- 55
lated kinases (ERKs) ERK1 and ERK2.) does not flow well.

Typos: vHCL vs HCL in the paragraph lines 73-81. More instead of most in line 256. Ranging vs ranged in 149.

Also additional references such as recent reviews (Yilmaz E, Chhina A, Nava VE, Aggarwal A. A Review on Splenic Diffuse Red Pulp Small B-Cell Lymphoma. Curr Oncol. 2021 Dec 6;28(6):5148-5154. doi: 10.3390/curroncol28060431. PMID: 34940070; PMCID: PMC8700110.) could be included for lines 67 or 82.

Author Response

Reviewer 3

This an interesting paper worth publication.  The english can be improved.

For example, this sentence (ase targets is activated: mitogen-activated protein kinase kinases (MEKs) MEK1 and MEK2, which phosphorylate extracellular signal-regulated kinases (ERKs) ERK1 and ERK2.) does not flow well.

Thank you for the remark.

English was corrected and if necessary, we can use MDPI editing service. We corrected in particular the underlined sentence above:

Lines 54-55: Downstream kinases activation by the mitogen-activated protein kinase kinases (MEKs) MEK1 and MEK2 that leads to the phosphorylation of extracellular signal-regulated kinases (ERKs) ERK1 and ERK2.

Typos: vHCL vs HCL in the paragraph lines 73-81. More instead of most in line 256. Ranging vs ranged in 149.

We modified.

Also additional references such as recent reviews (Yilmaz E, Chhina A, Nava VE, Aggarwal A. A Review on Splenic Diffuse Red Pulp Small B-Cell Lymphoma. Curr Oncol. 2021 Dec 6;28(6):5148-5154. doi: 10.3390/curroncol28060431. PMID: 34940070; PMCID: PMC8700110.) could be included for lines 67 or 82.

Again, many thanks to the reviewers. We added the reference.

vHCL vs HCL in the paragraph lines 73-81.

We can’t change the HCL immunological score by vHCL immunological score because it was defined as “HCL immunological score”. We changed the sentence to avoid confusion by: The immunological score is low: there is usually no expression of CD25 and CD200.

  • More instead of most in line 256

We changed the sentence by: CD11c was described in more than 95% of cases [11,14].

  • Ranging vs ranged in 149.

We changed the sentence by: Among the 82 patients, 63 were men and 19 were women, and ages ranging from 30 to 95 years.

And the more recent reference proposed was added line 82 (ref number 17).

Reviewer 4 Report

GENERAL COMMENTS

The aim of the present study is to distinguish HCL and HCL-like disorders.

The series is probably large from a clinical point of view, due to the rarity of Hairy Cell Leukemia, but it is rather small from a statistical point of view. HCL patients are 68, while the others are only 14 (5 vHCL/SDRPL and 9 HCL-like NOS, not otherwise specified). Hence the number of non-HCL patients is very small (n=14).

The Authors should be much more careful in their conclusion. For instance, when 26 adding CD26 to the immunologic HCL scoring system, specificity improves from 78.6% to 100%, but this is based on 14 patients.

Indeed, the Authors sometimes acknowledged the limited power of their study in the Discussion: Line 264: “Because of the small number of vHCL, we were not able to confirm CD79b” Lines 269-270: “In our study, CD38 expression was not associated 269 with a shorter TTNT and could be explained by the small size of the cohort …”.

In other words, the Authors should acknowledge that their study is preliminary and explorative.

MAJOR COMMENTS

1.Due to the limited sample size, the Authors could not divide their sample in a training set and a validation set, hence the sensitivity and specificity of their classification is somewhat optimistic, and this should be acknowledged in the Limitations section.

2.In the following sentence (lines 177-180): “The analysis of cHCL depending on expression of CD43, CD5, CD10, CD38 and CD4/CD8 T cells ratio did not show a significative difference in OS or PFS but a longer TTNT in case of CD23 positive expression (144.4 vs. 63.1 months) (Figure 2a and Figure S2)”, the Authors should acknowledge that “the difference was not significant (p=0.07), maybe for the low statistical power (n=15 and n=9, respectively)”.

3.Figure 2 is not well explained. For instance, black thin dotted lines are used to identify medians and this should be reported in the legend. Red areas and blue areas likely represent confidence intervals of survival curves. The reader can appreciate that confidence intervals are enormous. Red and blue areas should be skipped, as they represent an attempt to perform inference on survival, while the inference is actually not reliable due to the very low sample size (15+9 in panel A, and 13+5 in panel B). It would be better to simply report survival curves with descriptive purposes.

4.The Authors should better explain unsupervised analysis. Did unsupervised analysis consist in dimensionality reduction through Principal Component Analysis? Indeed, lines 200-202 seem to suggest that unsupervised analysis and PCA were different: “Unsupervised analysis AND principal component analysis (PCA) allowed the use of multiple continuous variables and reduced it to two dimensions to visualize samples associated with clusters”.

MINOR COMMENTS

5.As sample size is lower than 100, percentages should be rounded to integer values. For instance, 1/7 = 14.3%, round to 14%.

6.Line 177: “Significative difference” should be “significant difference.”

7.Variable factor map is indeed in the right upper corner of Figure 3, but it is in the centre of the map in Figure 4.

Author Response

Reviewer 4

The aim of the present study is to distinguish HCL and HCL-like disorders.

The series is probably large from a clinical point of view, due to the rarity of Hairy Cell Leukemia, but it is rather small from a statistical point of view. HCL patients are 68, while the others are only 14 (5 vHCL/SDRPL and 9 HCL-like NOS, not otherwise specified). Hence the number of non-HCL patients is very small (n=14).

The Authors should be much more careful in their conclusion. For instance, when 26 adding CD26 to the immunologic HCL scoring system, specificity improves from 78.6% to 100%, but this is based on 14 patients.

Indeed, the Authors sometimes acknowledged the limited power of their study in the Discussion: Line 264: “Because of the small number of vHCL, we were not able to confirm CD79b” Lines 269-270: “In our study, CD38 expression was not associated 269 with a shorter TTNT and could be explained by the small size of the cohort …”.

In other words, the Authors should acknowledge that their study is preliminary and explorative.

We agree with the comment and we added in the abstract:

However, these preliminary results have to be confirmed in a further study with a higher number of patients.

1.Due to the limited sample size, the Authors could not divide their sample in a training set and a validation set, hence the sensitivity and specificity of their classification is somewhat optimistic, and this should be acknowledged in the Limitations section.

You are totally right. A larger cohort would allow us to divide in a training and a validation set. To moderate our discussion and include this limitation, we corrected as follows:

Line 350: Due to a low number of patients, it was not possible to split the cohort in a training and a validation set. The classification must be validated in a prospective cohort.

2.In the following sentence (lines 177-180): “The analysis of cHCL depending on expression of CD43, CD5, CD10, CD38 and CD4/CD8 T cells ratio did not show a significative difference in OS or PFS but a longer TTNT in case of CD23 positive expression (144.4 vs. 63.1 months) (Figure 2a and Figure S2)”, the Authors should acknowledge that “the difference was not significant (p=0.07), maybe for the low statistical power (n=15 and n=9, respectively)”.

We also agree with that. We specified that the difference was not significant.

3.Figure 2 is not well explained. For instance, black thin dotted lines are used to identify medians and this should be reported in the legend. Red areas and blue areas likely represent confidence intervals of survival curves. The reader can appreciate that confidence intervals are enormous. Red and blue areas should be skipped, as they represent an attempt to perform inference on survival, while the inference is actually not reliable due to the very low sample size (15+9 in panel A, and 13+5 in panel B). It would be better to simply report survival curves with descriptive purposes.

Thank you for the comment. We clarified.

We modified the figures and removed the confidence intervals which complicate the understanding of the figure.

Line 212: Dotted lines represent median of TTN and PFS and red and blue areas the confidence of intervals of survivals curves.

4.The Authors should better explain unsupervised analysis. Did unsupervised analysis consist in dimensionality reduction through Principal Component Analysis? Indeed, lines 200-202 seem to suggest that unsupervised analysis and PCA were different: “Unsupervised analysis AND principal component analysis (PCA) allowed the use of multiple continuous variables and reduced it to two dimensions to visualize samples associated with clusters”.

We explained better.

Unsupervised analysis consists of dimensional reduction by PCA. We corrected the sentence, removed the risk of confusion as follows:

Line 238: Unsupervised analysis by principal component analysis (PCA) allowed the use of multiple continuous variables and reduced it to two dimensions to visualize samples associated with clusters.

  MINOR COMMENTS

5.As sample size is lower than 100, percentages should be rounded to integer values. For instance, 1/7 = 14.3%, round to 14%.

We changed: it is more appropriate.

6.Line 177: “Significative difference” should be “significant difference.”

We changed. Our apologies.

7.Variable factor map is indeed in the right upper corner of Figure 3, but it is in the centre of the map in Figure 4.

We harmonized.

The legend was corrected.

Line 274: Variable factors are shown in grey in the center.

Round 2

Reviewer 1 Report

The authors have addressed the reviewer's concerns.

Reviewer 2 Report

I agree with the Authors that HCL is a rare disease. For this, I invite them to involve other centers in order to achieve a number of cases adequate to test their diagnostic model.